# Well-Defined Pre-Catalysts in Amide and Ester Bond Activation

**DOI:** 10.3390/molecules24020215

**Published:** 2019-01-09

**Authors:** Sandeep R. Vemula, Michael R. Chhoun, Gregory R. Cook

**Affiliations:** Department of Chemistry and Biochemistry, North Dakota State University, Fargo, ND 58108–6050, USA; sandeepreddy.vemula@ndsu.edu (S.R.V.); michael.chhoun@ndsu.edu (M.R.C.)

**Keywords:** pre-catalysts, palladium catalysis, amide bond activation, ester bond activation, cross-coupling

## Abstract

Over the past few decades, transition metal catalysis has witnessed a rapid and extensive development. The discovery and development of cross-coupling reactions is considered to be one of the most important advancements in the field of organic synthesis. The design and synthesis of well-defined and bench-stable transition metal pre-catalysts provide a significant improvement over the current catalytic systems in cross-coupling reactions, avoiding excess use of expensive ligands and harsh conditions for the synthesis of pharmaceuticals, agrochemicals and materials. Among various well-defined pre-catalysts, the use of Pd(II)-NHC, particularly, provided new avenues to expand the scope of cross-coupling reactions incorporating unreactive electrophiles, such as amides and esters. The strong σ-donation and tunable steric bulk of NHC ligands in Pd-NHC complexes facilitate oxidative addition and reductive elimination steps enabling the cross-coupling of broad range of amides and esters using facile conditions contrary to the arduous conditions employed under traditional catalytic conditions. Owing to the favorable catalytic activity of Pd-NHC catalysts, a tremendous progress was made in their utilization for cross-coupling reactions via selective acyl C–X (X=N, O) bond cleavage. This review highlights the recent advances made in the utilization of well-defined pre-catalysts for C–C and C–N bond forming reactions via selective amide and ester bond cleavage.

## 1. Introduction

Transition metal-catalyzed cross-coupling reactions to form C–C and C–N bonds are a mainstay of organic synthesis for a wide range of academic and industrial applications [1,2,3,4,5,6]. Due to their wide applicability, these reactions have become a critical arsenal for synthetic chemists and have clearly changed retrosynthetic analysis of complex targets. Since their discovery in the late 1960s, palladium catalyzed cross-coupling reactions has been considerable and continues to be a focus of organometallic research [7,8,9,10,11,12,13,14]. The most active Pd catalysts for cross-coupling reactions involve the use of strong donor ligands to reach a high degree of efficiency. In fact, one of the major advancement in cross-coupling reactions is the synthesis and utilization of specialized electron-rich phosphines and *N*-heterocyclic carbenes (NHC) for the development of active catalytic systems expanding the substrate scope with lower catalyst loadings and milder conditions [15,16]. However, the monetary costs of these specialized ligands are often comparable to the Pd precursor. Therefore, the traditional route of addition of excess ligand for generating the active Pd(0) becomes unattractive [15,17]. Furthermore, in many cross-coupling reactions, the optimal Pd to ligand ratio is 1:1, with the active species proposed to be a monoligated Pd(0). Therefore, the use of well-defined Pd(II) pre-catalysts to facilitate cross-coupling reactions is highly desirable, as they can generate mono ligated active Pd(0) catalysts in solution. Since Herrmann reported that Pd-NHC complexes efficiently catalyzed Heck reaction [18], these complexes found a widespread use for various cross-coupling reactions incorporating previously unreactive coupling partners (Figure 1) [19].

Although various electrophiles are employed in cross-coupling reactions for the construction of C–C and C–N bonds, there is immense interest in increasing the substrate scope to include a wide range of cross-coupling partners [1,4,20]. In recent years, tremendous progress was made to incorporate stable, unreactive, carboxylic acid derivatives, such as amides and esters, in cross-coupling reactions to form ketones or amides [20,21]. The resonance, due to n_N (or) O_ →π*_C=O_ conjugation, makes them a complicated reacting partner in cross-coupling reactions, requiring a high activation energy for the N or O–C(O) bond scission. Destabilization strategies increasing the steric bulk on the amide nitrogen independently developed by the laboratories of Garg [22], Szostak [23], and Zou [24] provide a basis for the development of acyl cross-coupling of amides. Various Pd and Ni catalyst systems with phosphine and NHC ancillary ligands are shown to be effective in utilizing amides and esters as coupling partners in Suzuki-Miyaura coupling and Buchwald-Hartwig amination reactions. However, in this review, we will focus only on describing the recent advances made in the cross-coupling of amides and esters using well-defined pre-catalysts. The individual sections in this review are organized according to the pre-catalysts employed [Pd(allyl)(NHC)Cl, and Pd-PEPPSI] for cross-coupling of amides and esters. 

## 2. Pd(allyl)(NHC)Cl Pre-Catalysts in Suzuki-Miyaura Cross-Coupling of Amides and Esters

With a wide functional group tolerance and less sensitivity toward air and moisture, palladium is by far the most commonly used metal for catalysis of cross-coupling reactions. Among the various pre-catalysts developed in the past decade, there is considerable interest and improvement in pre-catalysts based on η^3^-allyl ligands. Aside from the inherent advantages associated with NHC ligands (i.e., strong σ-donation, tunability, sterics), Nolan demonstrated that η^3^-allyl and η^3^-cinnamyl pre-catalysts with bulky NHC ligands also exhibit high stability toward air and moisture allowing for easy storage and handling [15,25,26]. Furthermore, the commercial availability and ease of synthesis made them an important class of catalysts for reaction screening. The strong σ-donating nature of NHC facilitates the activation of stable and unreactive bonds. Specifically, the [Pd(cinnamyl)(L)Cl] scaffold (L = IPr* or IPr*OMe) was incorporated into several highly active catalysts for difficult cross-coupling reactions, such as the synthesis of tetra-*ortho*-substituted biaryls using Suzuki-Miyaura coupling [27], Buchwald-Hartwig reactions with secondary amines [28], as well as the use of amides and esters as electrophiles in Suzuki-Miyaura coupling and Buchwald-Hartwi aminations. Figure 2 lists selected and active Pd(allyl)NHC pre-catalysts that were employed for the cross-coupling of amides and esters [19].

### 2.1. Pd(allyl)(NHC)Cl Pre-Catalysts in Suzuki-Miyaura Cross-Coupling of Amides

Given the key role of the amide bond in nature, development of new methods to functionalize amides via C–N bond activation became an active area of research. Mechanistically, activation of the C–N amide bond proceeds through ground-state destabilization (twisting) of the amide bond by steric and/or electronic factors, allowing a facile insertion of a metal into the C–N bond, furnishing the acyl–metal intermediate facilitating coupling reactions [29,30,31]. Scheme 1 lists various substituents on amide nitrogen employed for the destabilization.

In 2015, the utilization of amide bond for the Suzuki-Miyaura cross-coupling was reported by Zou and coworkers employing *N*-phenyl-*N*-tosyl substituted amides [24]. The report showcased the effect of substituent electronics on the destabilization of amide resonance facilitating the metal insertion in to the amide C–N bond. Although the methodology showed good functional group tolerance, it suffered from high catalyst loadings and stringent reaction conditions (Scheme 2).

In the same year, Szostak and coworkers employed a rather different approach of sterically controlled amide bond destabilization using a variety of *N*-substituted amides and found the best results to be *N*-glutarimide [23]. The methodology provided inherent steric advantages for amide bond distortion, thereby providing a milder reaction conditions and high functional group tolerance (Scheme 3). But the utilization of large excess of expensive ligand (1:4 ratio of Pd(OAc)_2_ to PCy_3_HBF_4_), and applicability to only highly twisted *N*-glutarimide amides restricted the practical applications of the methodology. 

It was not until recently that a more general method incorporating all classes of amides under milder conditions was developed. In 2017, Szostak and coworkers reported the versatility of [Pd(IPr)(cinnamyl)Cl], a well-defined pre-catalyst in the Suzuki-Miyaura cross-coupling of amides under operationally simple conditions (Scheme 4) [23]. The methodology is highly significant, as it promoted the C–N amide bond activation of various amides, including *N*-glutarimide, *N*-Boc-carbamate, and *N*-toluenesulfonamide, under the same reaction conditions with similar yields of the ketone product. The method also showed high functional group tolerance for both reacting partners. The use of easily prepare and common protecting groups (*N*-Boc and *N*-Ts amides) for cross-coupling is appealing, as it brings the amide bond cross-coupling closer to general practical use. As stated previously, it was proposed that strong σ donation of the NHC facilitates oxidative addition, while flexible steric bulk around Pd promotes reductive elimination, triggering high reactivity even with less activated amide precursors.

While nonplanar amides were found to be very effective and well-documented for cross-coupling reactions by steric distortion, the resonance destabilization of corresponding planar amides was not well developed. To incorporate planar amides for cross-coupling reactions, Szostak and coworkers utilized *N*-acylpyrroles and *N*-acylpyrazoles in Suzuki-Miyaura cross-coupling reactions (Scheme 5) [32]. They found delocalization of N_lp_ into the π-electron system of the pyrrole/pyrazole ring to be sufficient to selectively insert palladium into the amide C–N bond in the absence of steric distortion. The extensive optimization emphasized the importance of well-defined [Pd(cinnamyl)(IPr)Cl] producing ketone in good yields compared to traces of product formation with the traditional catalyst/ancillary ligand system.

Compared to both traditional Pd/phosphine catalyst systems or Pd/ancillary NHC catalytic systems, well-defined [Pd(cinnamyl)(IPr)Cl] offers both practical advantages and high activity in cross-coupling reactions of amides. In fact, the high catalytic activity of [Pd(cinnamyl)(IPr)Cl] can be evident by its ability to convert *N*-alkyl-*N*-aryl amides to corresponding ketones. In 2017, Szostak and coworkers introduced *N*-methylaminopyrimidyl amides (MAPA) as highly reactive, electronically activated amides for C–N bond cleavage (Scheme 6) [33,34].

Interestingly, Suzuki-Miyaura cross-coupling of both the sterically distorted and/or electronically distorted amides were found to be reactive under similar reaction conditions, further demonstrating how the well-defined nature of a catalyst can enhance the overall catalytic activity and reactivity compared to traditional usage of excess ancillary ligand. This is highly beneficial, as it allows chemists to screen minimal conditions to optimize methodology for new classes of amides with similar distortion angles or resonance energies. In fact, Zeng and coworkers, on their quest to incorporate commercially available and inexpensive *N*-acylhydantoins as amide electrophiles, have also used similar conditions for the formation of ketones indicating the versatility of Pd(allyl)(NHC)Cl pre-catalysts with different substituted amides (Scheme 7) [35].

Although the reported methods to incorporate amide bonds in cross-coupling reactions is quite promising, the synthetic modifications to activate amide bond can impede overall synthetic utility. On the other hand, use of *N*,*N*-di-Boc-activated amides would be highly beneficial, allowing direct engagement of most ubiquitous primary amides. Szostak and coworkers developed a new catalytic system based on [Pd(IPr)(cinnamyl)Cl]/KF for the selective insertion of metal into acyl amide bond (cf. *N*-carbamate bond) [36]. The addition of 5.0 equiv of water was found to be critical for the cross-coupling of di-boc amides, as is evident from decreased yields or no reaction in the absence of water (Scheme 8). 

The high catalytic activity of Pd(IPr)(cinnamyl)Cl allowed for the cross-coupling to be conducted at ambient temperature with even lower catalyst loadings. The robust nature of Pd(IPr)(cinnamyl)Cl also allowed them for the gram scale conversion of primary amides to ketones with just 0.5 mol % of catalyst with a TON of 1220, further increasing the practicality of amide cross-coupling (Scheme 9) [36]. In fact, the high catalytic activity of Pd-IPr complexes also allowed Szostak and coworkers to use easily accessible *N*-acyl amides for the cross-coupling reaction with the highest reported TON (3500), compared to the corresponding traditional Pd-phosphine catalytic systems [37].

Recently, employing standard reaction conditions of amide Suzuki-Miyaura cross-coupling under well-defined Pd-NHC catalysis (i.e., Pd(IPr)(cinnamyl)Cl/K_2_CO_3_/THF), Szostak and coworkers also realized challenging C(sp^2^)-C(sp^3^) couplings using *N*,*N*-di-Boc-amides and B-sp^3^ alkyl reagents (Scheme 10) [38]. Various substituents on the amide bonds, such as *N*-glutarimide, *N*-Ph-*N*-Boc, and *N*-Ph-*N*-Ms were also found to be reactive under identical conditions albeit at a higher temperature.

### 2.2. Pd(η^3^-1-t-Bu-indenyl) (NHC)Cl Pre-Catalysts in Suzuki-Miyaura Cross-Coupling of Amides

The high reactivity of [Pd(allyl)(NHC)Cl] pre-catalysts stems from maintaining the optimal 1:1 Pd to ligand ratio and the fast reduction of Pd(II) to active Pd(0) [15]. It is now widely accepted that base mediated activation of [Pd(allyl)(NHC)Cl] pre-catalysts forms the active ligated Pd(0) catalyst in solution [34]. However, elegant studies by Hazari and coworkers observed a deleterious comproportionation pathway under catalytic conditions to form Pd(I) μ-allyl dimers (Scheme 11) [39,40]. Based on extensive mechanistic investigation, they proposed two strategies to develop even more active catalysts: (1) Increase the barrier to comproportionation between Pd (0) and Pd (II), and (2) develop systems that undergo faster activation so that all of Pd(II) is converted to Pd(0) before comproportionation. 

The mechanistic insights on the deactivation pathway led the Hazari group to discover Pd(η^3^-1-*t*-Bu-indenyl)(NHC)Cl (Hazari catalyst), a highly efficient pre-catalyst for cross-coupling reactions [41]. As a major advance in Pd-NHC precatalysts, the inability of (η^3^-1-*t*-Bu-indenyl) (NHC)Cl to generate unreactive Pd(I) dimers significantly improved its activity for cross-coupling reactions. In 2017, Szostak and coworkers utilized [Pd(indenyl)(IPr)Cl] for the development of Suzuki-Miyaura cross-coupling of *N*-Ph-*N*-Boc amides under otherwise standard conditions of amide cross-coupling reaction (K_2_CO_3_/THF, Scheme 12) [42]. The unprecedented reactivity of Hazari catalyst by suppressing the deactivation pathway led the amide cross-coupling reaction to occur at room temperature under glovebox-free conditions, increasing the operational simplicity and practicality of the reaction. This reaction is notable as the first example of Suzuki-Miyaura cross-coupling of an amide at room temperature with excellent yield of ketone product. The robust nature of the pre-catalyst allowed the Szostak group to develop the first direct one-pot activation/cross-coupling of secondary amides.

### 2.3. Pd(allyl)(NHC)Cl Pre-Catalysts in Suzuki-Miyaura Cross-Coupling of Esters

The ubiquitous nature of ester bonds attracted the interest of chemists for their synthetic manipulation into useful products. In recent years, the direct use of aromatic esters in cross-coupling reactions to form biaryls or ketones has been demonstrated as a promising area of research [20]. The selective cleavage of ester bonds offers significant advantages in multistep synthesis as they are robust to a range of reaction conditions and can only be activated under specific conditions. Although activation of aryl C–O bonds to form biaryls via decarbonylation is well-documented [43,44] the corresponding synthesis of ketones is still in its infancy. In 2017, Newman, Houk and coworkers reported the first-time utilization of aryl esters as acyl equivalents for the formation of ketones (Scheme 13) [45]. The exceptional bulkiness of NHC ligand on the catalyst hindered the decarbonylation step facilitating ketone formation over the well-known biaryl formation. 

Optimization studies demonstrated the importance of well-defined pre-catalysts for successful ketone formation (Table 1). When Pd catalysts in combination with excess ancillary ligand were used, very low yield of the product was obtained (Table 1, entry 1–4). On the contrary, when the metal to ligand ratio was decreased to the optimal 1:1 ratio, a substantial increase in the product formation was observed (Table 1, entry 5). Further improvement was reported when a preformed well-defined Pd(IPr)(cinnamyl)Cl was used as the catalyst, forming the cross-coupled product in 95% yield, highlighting the importance of well-defined pre-catalysts as compared to traditional catalyst/ligand system (Table 1, entry 6).

Very recently, as an improvement from their work on C(sp^2^)-C(sp^2^) cross-coupling of esters, the Newman group reported a C(sp^2^)-C(sp^3^) cross-coupling of aryl esters with alkyl-BBN to form aryl-alkyl ketones [46]. This reaction is particularly challenging, as alkyl boron reagents, in general, are reluctant to undergo transmetallation relative to their aryl counterparts and the intermediacy of alkylmetal species are prone to *β*-hydride elimination. However, the strong σ-donation and bulky nature of NHC in precatalysts, enabled the reaction to proceed to form ketones. On the contrary, when they employed Pd-ancillary phosphine catalyst conditions, biaryls were formed via decarbonylation and reductive elimination (Scheme 14). 

### 2.4. Pd(η^3^-1-t-Bu-indenyl) (NHC)Cl Pre-Catalysts in Suzuki-Miyaura Cross-Coupling of Esters

The Hazari catalyst, a highly active catalyst for amide bond cross-coupling, was also found to be very effective in Suzuki-Miyaura cross-coupling of aryl esters. In 2017, Szostak and coworkers reported the first room temperature Suzuki-Miyaura cross-coupling of esters by selective C–O bond activation to give biaryl ketones (Scheme 15) [42]. Interestingly, only a small variation in the amount of base from the conditions employed for amide cross-coupling enabled the cross-coupling of esters.

An advancement in the cross-coupling of aryl esters catalyzed by Pd(indenyl)(IPr)Cl was reported by Hazari and coworkers (Scheme 16) [47]. The use of potassium hydroxide base allowed them to develop even milder reaction conditions with lower catalyst loadings (1.0 mol %), and shorter reaction times (6 h to 16 h). 

## 3. Pd(allyl)(NHC)Cl Pre-Catalysts in Buchwald-Hartwig Amination of Amides and Esters

### 3.1. Pd(cinnamyl)(IPr)Cl Pre-Catalyst for the Transamidation of Amides

The excellent reactivity of Pd(cinnamyl)(IPr)Cl pre-catalysts in amide C–N activation to form acyl-metal species is highly significant, as it can potentially undergo cross-coupling reactions with any nucleophile. In 2017, Szostak and coworkers realized the acyl-metal reaction with other nucleophiles when they reported the first general method for Buchwald-Hartwig amination of acyl-metal species with amines under Pd-NHC catalysis (Scheme 17) [48]. The protocol offered advantages in efficient cross-coupling of both alkyl, aryl, and sterically hindered anilines. The method also showed a broad scope with respect to both the amine and amide components.

### 3.2. Pd(allyl)(IPr)Cl Pre-Catalyst for the Transamidation of Esters

Newman and coworkers reported the first example of Pd-NHC catalyzed amide bond formation directly from aryl esters and anilines. Similar to their observation in Suzuki coupling of aryl esters, a well-defined preassembled NHC-ligated catalyst was essential, as separated components resulted in reduced yields [45]. They found the use of a mild carbonate base and the presence of water was essential for higher conversion, forming the transamidated product in good yields (Scheme 18). The mildness of the protocol was showcased by the reaction of proline ester with anilines and the aminated product was formed with minimal loss of enantiopurity.

### 3.3. Pd(indenyl)(SIPr)Cl Pre-Catalyst for the Transamidation of Esters

Given the exceptional catalytic activity of Pd(indenyl)(IPr)(Cl) for Suzuki-Miyaura reactions involving aryl esters, Hazari and coworkers also explored Buchwald-Hartwig amination of aryl esters. Using 1 mol % of Pd(indenyl)(SIPr)(Cl) (SIPr = 1,3-bis(2,6-diisopropylphenyl)imidazolidin-2-ylidene) as the pre-catalyst, they were able to couple phenyl benzoate and aniline in essentially quantitative yield at 40 °C, using a 4:1 H_2_O/THF solvent mixture (Scheme 19) [47]. They also found that the coupling was highly sensitive to the ligand used, with SIPr affording the best yields, whereas other NHC or phosphine ligands were ineffective.

## 4. Pd-PEPPSI Pre-Catalysts in the Suzuki-Miyaura Cross-Coupling of Amides and Esters

The last two decades saw tremendous growth in the development of highly active, generally applicable, and functional group-tolerant catalytic systems employing NHC ligands. One highly active well-defined pre-catalyst system is Pd-PEPPSI (pyridine-enhanced pre-catalyst preparation, stabilization and initiation), developed by the Organ group [49]. As Figure 1 illustrates, Pd-PEPPSI catalysts found their use in various cross-coupling reactions, such as Suzuki-Miyaura, Negishi, Kumada-Tamao-Corriu, and Buchwald-Hartwig aminations [50]. As with all the Pd-NHC precatalysts, Pd-PEPPSI has two major components: The bulky NHC ligand, which contains strong σ-donor properties which help promote the oxidative addition, and the sterics aid reductive elimination. The second component, the pyridine, often referred as a “throw away ligand,” aids not only in the synthesis of pre-catalyst, but also increases the stability of the isolated complexes (Figure 3). The easy synthesis, high stability toward air and moisture and the exceptional reactivity of Pd-PEPPSI complexes led synthetic chemists to utilize these pre-catalysts in the cross-coupling of difficult and unreactive electrophiles, such as amides and esters (*vide infra*). 

### 4.1. Pd-PEPPSI Pre-Catalysts in the Suzuki-Miyaura Cross-Coupling of Amides 

Szostak and coworkers were the first to report the use of Pd-PEPPSI pre-catalysts in Suzuki-Miyaura cross-coupling of amides [51]. In agreement with high catalytic activity of in situ generated Pd-NHC complexes, identical reaction conditions could be employed for the cross-coupling of various destabilized amides (Scheme 20).

In accordance with the amide C–N bond destabilization, kinetic studies with *N*-glutarimide, *N*-Ph-*N*-Ts, and *N*-Ph-*N*-Boc amides showed similar reactivity to their cross-coupling reactivity scale with *N*-glutarimide being the fastest and *N*-Ph-*N*-Ts being the slowest [52]. The kinetic data between Pd-PEPPSI and Pd(IPr)(cinnamyl)Cl precatalysts under identical conditions suggested tha while the reaction rates of *N*-glutarimide and *N*-Ph-*N*-Boc amides were higher with Pd-PEPPSI, *N*-Ph-*N*-Ts amides are shown to have faster reaction rates under Pd(IPr)(cinnamyl)Cl catalysis (Figure 4) [19]. The data is highly beneficial, as it provides a general idea for chemists to choose from both the catalyst systems for different type of amides employed in cross-coupling reactions.

Very recently, Zou and coworkers reported the Suzuki-Miyaura cross-coupling of less reactive *N*-alkyl-*N*-Ts amides with diarylborinic acids under Pd-PEPPSI catalysis (Scheme 21) [53]. The use of stable diarylborinic acids was advantageous, as it extended nucleophile counterparts in cross-coupling reactions. Under otherwise standard reaction conditions of amide cross-coupling reactions, they were successful in cross-coupling less reactive *N*-Me-*N*-Ts amides to form ketone products. Notably, no reaction ensued under Pd/phosphine catalysis and only low Pd-PEPPSI catalyst loadings (1.0 mol %) were needed, highlighting the high activity of Pd-NHC pre-catalysts. The protocol also offered a broad functional group tolerance, with both electron donating and electron withdrawing groups on both coupling partners. 

Zou and coworkers further advanced the Suzuki-Miyaura coupling of amides to form alkyl ketones using trialkylboranes as coupling partners under Pd-PEPPSI catalysis [54]. Although Pd/phosphine catalysis proved ineffective, 5 mol % of Pd-PEPPSI was very effective in transforming *N*-Me-*N*-Ts amides to form alkyl ketones (Scheme 22). Unlike the high-order arylboron compounds, in which all the aryl groups react effectively, only one of the three primary alkyl groups in alkylboranes could be used as alkyl source for the acyl alkylation.

Wang, Liu, and coworkers recently reported a new series of Pd-NHC pre-catalysts using various *N*-heterocycles [55] as “throwaway ligands” with benzothiazole being the most effective for the Suzuki-Miyaura cross coupling of *N*-succinimide amides (Scheme 23) [56]. Change in the “throw-away ligand” was found to have a profound effect on the overall yield of the reaction, with benzothiazole ligand outperforming traditional 3-chloro pyridine ligand. 

### 4.2. Pd-PEPPSI Pre-Catalysts in the Suzuki-Miyaura Cross-Coupling of Esters

Pd-PEPPSI pre-catalysts were also found to be effective in catalyzing the cross-coupling of esters. In most cases, the reactivity was comparable to that of [Pd(IPr)(cinnamyl)Cl]. Szostak and coworkers reported a direct Suzuki-Miyaura cross-coupling of aryl esters under Pd-PEPPSI catalysis and found that the rates were similar to those reported for the cross-coupling of *N*-Ph/Boc, and *N*-Ph/Ts amides under identical reaction conditions (Scheme 24) [57]. 

To further advance cross-coupling of esters, and to find a more general method, Szostak and coworkers reported a water assisted Suzuki-Miyaura cross-coupling of aryl esters at room temperature (Scheme 25) [58]. They demonstrated that the addition of water (5.0 equiv) was able to facilitate the reaction under very mild reaction conditions with only 1 mol % catalyst loading and at room temperature. 

To elucidate the role of water additive, they subjected the Pd-PEPPSI pre-catalyst under optimized reaction conditions without the coupling partners to observe the formation of hydroxide dimer, [Pd(μ-OH)Cl(IPr)]_2_ in appreciable 32% yield (Scheme 26). 

They also performed kinetic studies to probe the catalytic activity of preformed dimer and observed faster reaction rates compared to the Pd-PEPPSI conditions. Notably, negligible product formation in the absence of water additive indicated the importance of Pd-hydroxide dimer formation prior to the reduction of Pd(II) to active Pd(0) species (Figure 5). This protocol has also allowed the first achievement of a TON > 1000 in the Suzuki-Miyaura cross-coupling of aryl esters.

Very recently, while preparing this manuscript, Szostak and coworkers reported Suzuki-Miyaura cross-coupling of esters using pentafluorophenyl substituents. Although the reported method was primarily focused on the utilization of traditional Pd-ancillary phosphine catalysis, the high catalytic activity of Pd-PEPPSI catalysts allowed them to use even milder conditions with better yields [59].

## 5. Pd−PEPPSI Pre-Catalysts in Buchwald–Hartwig Amidation of Esters and Amides 

Szostak and Shi also reported the first Pd-PEPPSI catalyzed Buchwald-Hartwig amination of phenyl esters and activated amides. In their work, they reported the chemo-selective acyl C–O/C–N amination with anilines using the Pd-PEPPSI pre-catalyst [60]. Phenyl esters, *N*-Ph-*N*-Boc, and *N*-Ph-*N*-Ts amides were successfully converted to amides with excellent yield. However, conditions reported were harsher than conditions reported for ketone synthesis (Scheme 27).

## 6. Palladium(II)/N-Heterocyclic Carbene-Catalyzed Direct C–H Acylation of Heteroarenes with N-Acylsaccharins

Following up on the work from the Szostak group, Gandhi and coworkers reported the use of pyrene based Pd-PEPPSI catalysts (Figure 6) with wingtip substituents, such as Ar/alkyl and alkyl groups for the cross-coupling of *N*-acylsaccharin amides with azoles and azole derivatives [61].

*N*-acyl saccharin, a commonly used twisted amide, was used to accomplish the difficult cross coupling reactions incorporating amide bond cleavage and C-H activation reactions. The work focused on the cross-coupling of *N*-functionalized saccharins with benzoxazole and its derivatives through a C-H activation process (Scheme 28). Not surprisingly, other activated amides such as *N*-aryl-*N*-Ts were found ineffective, resulting in no product.

## 7. Conclusions 

In summary, the use of well-defined Pd(II)-NHC pre-catalysts offers significant advantages over the traditional approach of adding extra ligand to standard Pd catalysts. In the past few years, these catalysts were found to be exceptional in enabling the cross-coupling reactions of amides and esters via C–N(O) bond activation to form ketones and amides. The easy preparation, commercial availability, and stability toward air and moisture of the pre-catalysts facilitate the development of operationally simple and practical cross-coupling methods. The strong σ-bond donating nature of NHC ligands increases the reactivity of Pd center toward oxidative addition of difficult electrophiles, such as amides and esters, increasing the scope and generality of the cross-coupling reactions. Further exciting developments are expected in this relatively new area of employing amides and esters as acyl electrophiles, with recent breakthroughs enabled by Pd(II)-NHC catalysis.

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
