# Peer review of "Well-Defined Pre-Catalysts in Amide and Ester Bond Activation"

_molecules, 2019, doi:10.3390/molecules24020215_

Round 1

Reviewer 1 Report

Cook et al. Report a nice revision of, mainly, NHC-M complexes and their use as catalysts precursors using amide and esters in cross coupling reactions. In general, the manuscript is amenable for reading and comprehensive in a theme that is of current interest in both organic and organometallic chemistry. The references are fine both in number and current, however a couple of key references must be added before the paper is accepted (1) H. Valdés, D. Canseco-González, J. M. Germán-Acacio, D. Morales-Morales. J. Organomet. Chem. 2018, 867, 51-54. 2) D. Morales-Morales (Ed.), Pincer Compounds Chemistry and Applications, Elsevier, 2018.). Other than that, I believe that this manuscript would be of interest and welcome by the reading community of Molecules.

Author Response

Thank you for the kind feedback. We have included the suggested reference (number 16) and fixed other references and typos in the document. 

Reviewer 2 Report

This review deals with Suzuki-Miyaura coupling and Buchwald-Hartwig amination of amides and esters with Pd(II)-NHC precatalysts. This is an important area in Pd-catalyzed cross-coupling reactions. This review is well organized and will be useful for readers in the research field of Pd catalysis. Thus, I recommend this review for publication in Molecules, although some minor revisions are required as shown below.

(1) Scheme 2: “Pd(PCy3)Cl2” should be “Pd(PCy3)2Cl2”.

(2) Schemes 4, 5, and 6: “Pd(IPr)(cin)Cl” should be “Pd(IPr)(cinnamyl)Cl”, as shown in Schemes 8-10.

(3) Scheme 8, Caption: “cinamyl” should be “cinnamyl”.

(4) Page 6, Line 170: “cinamyl” should be “cinnamyl”.

(5) Scheme 10, Caption: “cinamyl” should be “cinnamyl”.

(6) Scheme 11: “IHPr” in the less reactive form should be “IPr”.

(7) Page 11, Line 312: “(IPr(cinnamyl)” should be “(IPr)(cinnamyl)”.

(8) Scheme 23: The curved bond between the two nitrogen atoms of the NHC is missing.

(9) Figure 6: The curved bond between the two nitrogen atoms of the NHC is missing.

Author Response

Thank you for the kind suggestions. We have made all of the minor edits suggested by the reviewer.